# Technical note: Updated parameterization of the reactive uptake of glyoxal and methylglyoxal by atmospheric aerosols and cloud droplets

Leah A. Curry, William G. Tsui, V. Faye McNeill

Department of Chemical Engineering, Columbia University, New York, NY, 10027, USA

*Correspondence to*: V. Faye McNeill (vfm2103@columbia.edu)

**Abstract**

We present updated recommendations for the reactive uptake coefficients for glyoxal and methylglyoxal uptake to aqueous aerosol particles and cloud droplets. The particle and droplet types considered were based on definitions in GEOS-Chem v11, but the approach is general. Liquid maritime and continental cloud droplets were considered. Aerosol types include sea salt

(fine and coarse) with varying relative humidity and particle size, and sulfate/nitrate/ammonium as a function of relative humidity and particle composition. We take into account salting effects, aerosol thermodynamics, mass transfer, and irreversible reaction of the organic species with OH in the aqueous phase. The new recommended values for the reactive uptake coefficients in most cases are lower than those currently used in large-scale models, such as GEOS-Chem. We expect application of these parameterizations will result in improved representation of aqueous secondary organic aerosol formation

in atmospheric chemistry models.

## 1 Introduction

The uptake and reaction of water soluble volatile organic compounds (VOCs) in cloud droplets and aerosol liquid water is likely a significant source of secondary organic aerosol (SOA) material (Carlton et al., 2008; Fu et al., 2009, 2008; McNeill, 2015; McNeill et al., 2012). These processes may be referred to, collectively, as aqueous SOA (or aqSOA) formation.

20        Glyoxal (CHOCHO, GLYX) and methylglyoxal ($CH_3C(O)CHO$, MGLY) are both atmospherically abundant gas-phase oxidation products of multiple VOC precursors, including isoprene and toluene. Both GLYX and MGLY are water-soluble, GLYX more so than MGLY (Betterton and Hoffmann, 1988; Zhou and Mopper, 1990). Taking into account salting effects, the effective Henry's Law constant for GLYX in aerosol water can be several orders of magnitude higher than that of MGLY, depending on the aerosol ionic content (Kampf et al., 2013; Waxman et al., 2015). As $\alpha$-dicarbonyl species, GLYX

and MGLY exhibit similar aqueous-phase chemistry: they undergo reversible hydration and self-oligomerization (Ervens and Volkamer, 2010; Hastings et al., 2005; Sareen et al., 2010; Shapiro et al., 2009), they can be oxidized by aqueous-phase radicals to form organic acids or organosulfates (Carlton et al., 2007; Lim et al., 2013; Perri et al., 2010; Schaefer et al., 2012, 2015), and they can react with nitrogen-containing species to form brown carbon (De Haan et al., 2018; Lee et al., 2013; Maxut et al.,

2015; Nozière et al., 2009; Powelson et al., 2014; Sareen et al., 2010; Schwier et al., 2010; Shapiro et al., 2009; Yu et al., 2011a).

GLYX and MGLY received significant attention in the atmospheric chemistry modelling community (Carlton et al., 2008; Fu et al., 2009, 2008) following early experimental demonstrations of their potential significance as aqSOA precursors (Carlton et al., 2007; Hastings et al., 2005; Kroll et al., 2005; Loeffler et al., 2006). Fu and co-workers predicted that uptake of GLYX and MGLY to low-level clouds was a significant source of organic aerosol over North America, with MGLY producing more than three times more SOA than GLYX (Fu et al., 2009). Carlton et al. (2008) found that including in-cloud aqSOA production by GLYX in CMAQ improved agreement with aircraft observations.

Since these initial studies, more information has become available regarding the gas-particle partitioning of glyoxal and methylglyoxal (Ip et al., 2009; Kampf et al., 2013; Waxman et al., 2015; Yu et al., 2011a) and their chemical processing in the aqueous phase, allowing a refinement of their representation in models.

Here, we calculate reactive uptake coefficients for glyoxal and methylglyoxal for several cloud and aerosol types for application in large-scale atmospheric chemistry modelling. We take into account salting effects, aerosol thermodynamics, mass transfer considerations, and aqueous phase chemical kinetics. We base our calculations on the cloud and aerosol types used in GEOS-Chem v11, so these recommendations can be applied directly to that model, but the approach is general.

## 2 Methods and Data

Following Hanson et al. (1994), the reactive uptake coefficient, $\gamma$, is calculated according to:

$$\frac{1}{\gamma} = \frac{1}{\alpha} + \frac{\omega}{4H^*\mathbb{R}T\sqrt{k^I D_{aq}}}\left(\frac{1}{cothq - 1/q}\right) \qquad (1)$$

where $\alpha$ is the mass accommodation coefficient, $\omega$ is the gas-phase thermal velocity of the organic species, $H^*$ is the effective Henry's Law constant (Schwartz, 1986), $\mathbb{R}$ is the universal gas constant, T is temperature in Kelvin, $k^I$ is the first order aqueous loss rate, and $D_{aq}$ is the aqueous-phase diffusion coefficient for the organic. Particle radius, $R_p$, and in-particle diffusion limitations are taken into account through the parameter $q = R_p/l$, where $l$ is the diffuso-reactive length:

$$l = \left(\frac{D_{aq}}{k^I}\right)^{1/2} \qquad (2)$$

The aqueous-phase diffusion coefficient used for both GLYX and MGLY was $D_{aq} = 10^{-9}$ m$^2$/s. $D_{aq}$ does not vary much for small species, and this value is typical for small organics (Bird et al., 2006). The mass accommodation coefficient used was $\alpha$ = 0.02. This value of $\alpha$ is an estimate based on the assumption that $\alpha$ values for GLYX and MGLY are similar to that of

formaldehyde uptake to water (Jayne et al., 1992). Following equation (1), the calculation is insensitive to within 10% for a 50% variation in $\alpha$ for values of $\gamma < 10^{-3}$.

***Particle types and composition.*** Although the approach described here is general, we applied it to the liquid cloud and aerosol particle types in GEOS-Chem v11. A complete listing can be found in the Supporting Information. Briefly, we considered marine and remote continental liquid cloud droplets, coarse and fine sea salt aerosol particles as a function of relative humidity, and sulfate/nitrate/ammonium (SNA) aerosols as a function of relative humidity and composition. Calculated results for $\gamma_{GLY}$ and $\gamma_{MGLY}$ as a function of S:A and S:N, and calculated pH are available in the Supporting Information. Sea salt aerosols are assumed to be composed of 100% NaCl. The Windows stand-alone executable for ISORROPIA-II (Fountoukis and Nenes, 2007) was used in forward mode to calculate the equilibrium inorganic ion composition of the aerosols, in order to calculate the Henry's constant. The temperature was held constant at 280 K and calculations were performed for each of the desired relative humilities (99, 95, 90, 80, 70, 50 and 0%). Solid formation was suppressed (metastable mode). For the SNA aerosols, the amount of $NO_3^-$ was held constant at 2 $\mu$mol/m$^3$ air, while the $SO_4^{-2}$ and $NH_3$ amounts were each varied from 1-8 $\mu$mol/m$^3$.

***Aqueous-phase reaction.*** The formulation in eq (1) and (2) describes uptake due to irreversible aqueous-phase loss processes only. Based on our previous analysis of the system using the multiphase photochemical box model GAMMA, the dominant irreversible atmospheric aqueous-phase reactive process for GLYX and MGLY is reaction with OH (McNeill, 2015; McNeill et al., 2012). This reaction is the initiation step for most radical-based chemistry of GLYX and MGLY in the atmospheric aqueous phase, including organic acid formation and organosulfate formation (McNeill et al., 2012; Perri et al., 2010). Other irreversible loss processes, such as imidazole formation, occur on much longer timescales (Teich et al., 2016; Yu et al., 2011). Therefore, the aqueous loss is represented by the pseudo first order rate constant for the reaction between the organic species of choice and OH (i.e., $k^I = k_{OH}[OH]$). Reversible reactive processes, e.g. spontaneous hydration and self-oligomerization of glyoxal and methylglyoxal, which substantially promote uptake of GLYX and MGLY to the aqueous phase, may be taken into account by the use of an effective Henry's Law constant (McNeill et al., 2012). However, we note that the form of eq. 1 implies no uptake (reversible or irreversible) in the absence of OH.

Considerable uncertainty exists in the aqueous concentration of OH in cloudwater and especially aerosol particles. In order to calculate $k^I$, we use OH concentrations for maritime and remote continental clouds and aerosols following Herrmann et al. (2010) (Table 1). They reported a range of [OH] for each scenario calculated using the CAPRAM 3.0 model. This [OH] range was used to calculate the uncertainty in $\gamma$.

**Table 1. Mean values and range of in-particle hydroxyl radical concentrations, as reported by Herrmann et al., (2010).**

| Cloud/aerosol type | Mean [OH] (M) | Max [OH] (M) | Min [OH] (M) |
|---|---|---|---|
| Maritime aerosols | $10^{-13}$ | $3.3 \times 10^{-12}$ | $4.6 \times 10^{-15}$ |
| Remote aerosols | $3.0 \times 10^{-12}$ | $8 \times 10^{-12}$ | $5.5 \times 10^{-14}$ |
| Maritime clouds | $2.0 \times 10^{-12}$ | $5.3 \times 10^{-12}$ | $3.8 \times 10^{-14}$ |
| Remote clouds | $2.2 \times 10^{-14}$ | $6.9 \times 10^{-14}$ | $4.8 \times 10^{-15}$ |

*Calculating the Henry's constant.* The solubility of glyoxal and methylglyoxal in aqueous solutions depends on the salt content (Ip et al., 2009; Kampf et al., 2013; Waxman et al., 2015; Yu et al., 2011a). Glyoxal becomes more soluble with increasing salt concentration (i.e., it exhibits "salting in"), whereas the opposite is true for methylglyoxal (it "salts out"). Therefore, the Henry's constants for glyoxal and methylglyoxal are a function of particle type and liquid water content (and therefore relative humidity).

10         The Henry's constants are calculated for sea salt aerosols following Waxman et al. (2015) using the equation:

$$\log\left(\frac{K_{H,w}}{K_{H,NaCl}}\right) = K_{s,NaCl}c_{NaCl} \qquad (3)$$

where $K_{H,w}$ is the Henry's constant for pure water, $K_{H,NaCl}$ is the Henry's constant for the salt-containing aerosol, $c_{NaCl}$ is the

NaCl concentration in molality as calculated using ISORROPIA-II, and and $K_{s,NaCl}$ is the salting constant (Table 2). Note that the $K_H$ values are effective Henry's constants, which account for hydration of the carbonyl species upon uptake. Waxman and coworkers showed that salting constants were additive for a mixed $(NH_4)_2SO_4/NH_4NO_3$ system, following:

$$\log\left(\frac{K_{H,w}}{K_{H,salt}}\right) = K_{s,(NH4)2SO4}c_{(NH4)2SO4} + K_{s,NH4NO3}c_{NH4NO3} \qquad (4)$$


where $K_{H,salt}$ is the Henry's constant for the salt mixture, $c_{(NH4)2SO4}$ and $c_{NH4NO3}$ are the concentrations in molality and $K_{s,(NH4)2SO4}$ and $K_{s,NH4NO3}$ are the salting constants. The sum of sulfate and bisulfate was used to calculate $c_{(NH4)2SO4}$.

    For cloud droplets, $K_{H,\,w}$ is used due to the low ion concentrations in cloudwater (Ervens, 2015; McNeill, 2015).


**Table 2. Reaction and mass transfer parameters**

| Species | $k_{OH}$ (M$^{-1}$ s$^{-1}$) | $K_{H,w}$ (M atm$^{-1}$) | $K_{s, NaCl}$ [1/m] | $K_{s, (NH4)2SO4}$ [1/m] | $K_{s, NH4NO3}$ [1/m] |
|---------|------------------------------|--------------------------|----------------------|---------------------------|-----------------------|
| Glyoxal | $1.1 \times 10^{9}$ [a] | $3.5 \times 10^{5}$ [c] | -0.10 [e] | -0.24 [d] | -0.07 [e] |
| Methylglyoxal | $7 \times 10^{8}$ [b] | $3.71 \times 10^{3}$ [c] | 0.06 [e] | 0.16 [e] | 0.075 [e] |

a (Schaefer et al., 2015)
b (Schaefer et al., 2012)
c (Betterton and Hoffmann, 1988)
d (Kampf et al., 2013)
e (Waxman et al., 2015)

*Statistical analysis and parameter estimation*. The calculated reactive uptake coefficient, γ, for MGLY and GLYX and each
particle type was parameterized as a function of relative humidity via weighted least squares regression. Assuming that the errors in the reactive uptake coefficients are log-normally distributed, a covariance matrix for the model parameters was calculated based on the mean square errors of the data and the design matrix of the linear regression. The standard deviations of the model parameters were then determined from the diagonal of the covariance matrix (Aster et al., 2005). Student's t-tests were then performed on each model parameter for the hypothesis that the model parameter in question is equivalent to zero in
order to assess the necessity of each parameter. The nonzero model parameter was kept for t-tests in which there was at least 98% confidence that the hypothesis of the model parameter being zero could be rejected.

## 3 Results and Discussion

The reactive uptake coefficient, γ, was calculated for MGLY and GLYX as a function of [OH], RH, particle size, and in the case of SNA aerosol, particle composition. Calculated values of γ varied over several orders of magnitude. In most cases these
values are lower than those previously used to model reactive uptake of these species in large-scale models.

### 3.1 Liquid cloud droplets.
The results for marine and remote continental cloud droplets are shown in Table 3 with the mean value and error bars given. The uncertainty reflects the uncertainty in [OH]. $\gamma_{MGLY}$ is lower than $\gamma_{GLYX}$ by a factor of roughly 100 in each case, consistent
with its lower $K_{H,w}$ and $k_{OH}$.

**Table 3. Recommended γ for liquid cloud droplets.** Cloud types and size as defined in GEOS-Chem v11.

| Cloud type | $R_{eff}$ (μm) | $\gamma_{GLYX}$ | $\gamma_{MGLY}$ |
|---|---|---|---|
| **Marine** | 10 | $7.5\times10^{-4}$ ($+0.001$,$-7.4\times10^{-4}$) | $5.7\times10^{-6}$ ($+9.4\times10^{-6}$,$-5.6\times10^{-6}$) |
| **Remote continental** | 6 | $4.3\times10^{-6}$ ($+9.2\times10^{-6}$,$-3.4\times10^{-6}$) | $3.2\times10^{-8}$ ($+6.7\times10^{-8}$,$-2.5\times10^{-8}$) |

### *3.2 Aerosols*.

For aerosols, the reactive uptake coefficients were found to vary significantly with RH due to salting effects. Figure 1 shows calculated values of $\gamma_{GLYX}$ for the three particle types as a function of RH. The range of uncertainty in the calculated values, indicated by the red shading, is due to the uncertainty in [OH] (Table 1), and in the case of SNA aerosols, variations due to the different aerosol compositions tested. The black lines indicate the weighted least squares fit to the data, and the grey lines indicate the confidence interval for the fit. The average values and the results of the least squares fits are summarized in Table 4.

$\gamma_{GLYX}$ decreases with increasing RH, due to salting in. Therefore, the maximum $\gamma_{GLYX}$ ($10^{-2}$ for SNA, 50% RH) exceeds the dilute (cloudwater) case. It also exceeds the general case used by Fu et al. (2008) ($\gamma_{GLYX, Fu} = 2.9\times10^{-3}$), which was based on the experimental observations of Liggio et al. (2005) for $(NH_4)_2SO_4$ aerosols at 55% RH. The parameterization presented in Table 4 yields $\gamma_{GLY} = 3.6\times10^{-3}$ at 55% RH.

In the case of coarse sea salt, in-particle diffusion limitations led to smaller $\gamma_{GLYX}$ at the mean [OH] than at the minimum [OH] for some RH values. The scatter in the calculated $\gamma_{GLYX}$ led to a low-confidence result from the weighted least squares regression. For this reason, we recommend use of the error-weighted average $\gamma_{GLYX}$ value in lieu of a parameterization (Table 4).

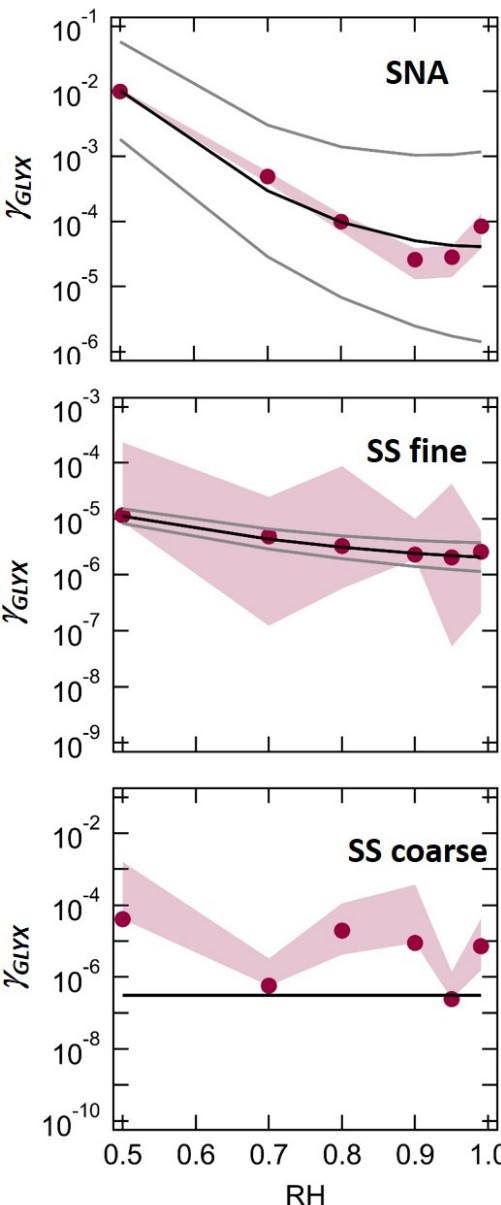

**Figure 1.** Calculated reactive uptake coefficients for uptake of glyoxal to sulfate/nitrate/ammonium (SNA) aerosols, and fine and coarse sea salt (SS) aerosols, as defined in GEOS-Chem v11. Red shading indicates the uncertainty in $\gamma_{GLYX}$. The black lines show the results of weighted least squares regression, with the confidence intervals in grey.

**Table 4. Summary of γ$_{GLYX}$ recommendations for aerosols.** Aerosol types and specifications as defined in GEOS-Chem v11.

| Aerosol type | γ$_{GLYX}$ average value | γ$_{GLYX}$ Parameterization<br>x: RH as fraction |
|---|---|---|
| SNA | 1.0(±0.1)×10$^{-2}$ (RH = 50%)<br>4.9(±1.0)×10$^{-4}$ (RH = 70%)<br>1.0(±0.3)×10$^{-4}$ (RH = 80%)<br>2.6(±1.3)×10$^{-5}$ (RH = 90%)<br>2.8(±1.4)×10$^{-5}$ (RH = 95%)<br>8.5(±4.6)×10$^{-5}$ (RH = 99%) | $\gamma = \exp(a + bx + cx^2)$<br><br>$a = 12.1\ (\pm 0.6)$<br>$b = -44.5\ (\pm 1.7)$<br>$c = 22.3\ (\pm 1.1)$<br>conf = 0.9997 |
| Sea salt (fine) | 2.6×10$^{-6}$ (+0.04,-2.6×10$^{-6}$) | $\gamma = \exp(a + bx + cx^2)$<br>$a = -7.5\ (\pm 0.1)$<br>$b = -10.0\ (\pm 0.3)$<br>$c = 4.4\ (\pm 0.2)$<br>conf = 0.9998 |
| Sea salt (coarse mode) | 4.8×10$^{-7}$ (+0.013,-4.8×10$^{-7}$) | Average value recommended |

Figure 2 shows calculated values of γ$_{MGLY}$ as a function of RH. The average values and the results of the least squares fits are summarized in Table 5. In contrast to glyoxal, methylglyoxal salts out, so γ$_{MGLY}$ increases with increasing RH. All calculated values ($10^{-10} < $ γ$_{MGLY} < 10^{-6}$) are much smaller than the general case used by Fu et al. (2008) (γ$_{MGLY, Fu} = 2.9 \times 10^{-3}$). Those investigators had assumed that the reactive uptake coefficient for methylglyoxal would be equal to that for glyoxal as measured

10  by Liggio et al. (2005).

Similar to the glyoxal case, the variability in γ$_{MGLY}$ for the coarse mode sea salt aerosols due to in-particle diffusion limitations led to a low-confidence weighted least squares fit. The error-weighted average value is recommended.

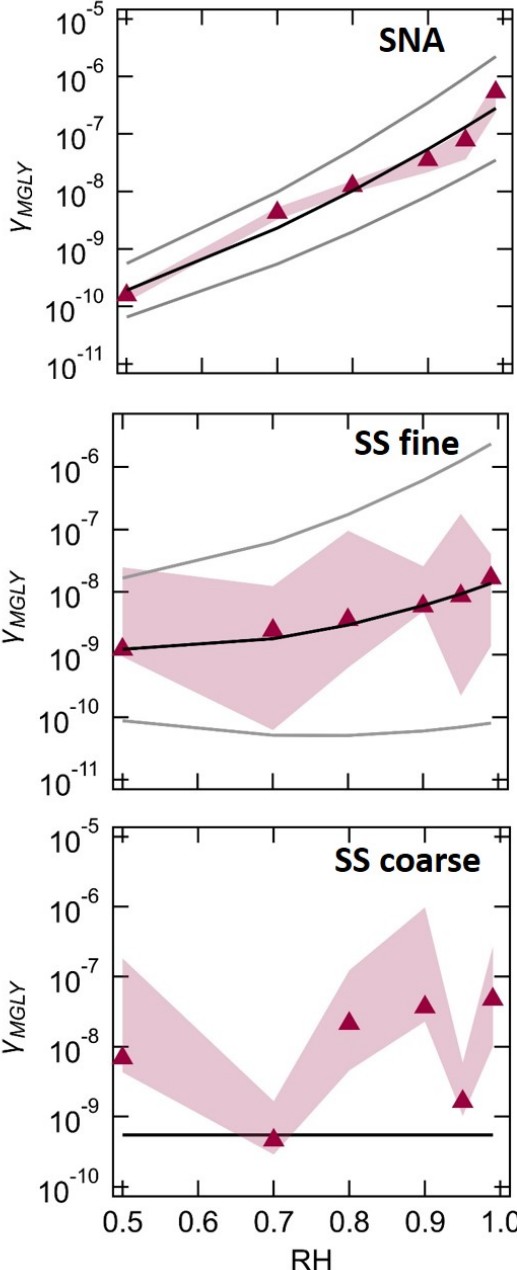

**Figure 2.** Calculated reactive uptake coefficients for uptake of methylglyoxal to sulfate/nitrate/ammonium (SNA) and sea salt (SS) aerosols. See text for details.

**Table 5. Summary of $\gamma_{MGLY}$ recommendations for aerosols.**

| Aerosol type | $\gamma_{MGLY}$ average value | $\gamma_{MGLY}$ Parameterization x: RH as fraction |
|---|---|---|
| SNA | $1.6(\pm0.4)\times10^{-10}$ (RH = 50%) $4.3(\pm1.0)\times10^{-9}$ (RH = 70%) $1.3(\pm0.3)\times10^{-8}$ (RH = 80%) $3.5(\pm1.5)\times10^{-8}$ (RH = 90%) $7.7(\pm4.1)\times10^{-8}$ (RH = 95%) $5.3(\pm2.9)\times10^{-7}$ (RH = 99%) | $\gamma = \exp(a + bx + cx^2)$ $a = -25.7\ (\pm 0.4)$ $b = 2.5\ (\pm 1.0)$ $c = 8.3\ (\pm 0.7)$ conf = 0.9990 |
| Sea salt (fine) | $6.5\ (\pm1.3) \times 10^{-9}$ | $\gamma = \exp(a + bx + cx^2)$ $a = -17.9\ (\pm 0.9)$ $b = -10.4\ (\pm 2.6)$ $c = 10.3\ (\pm 1.7)$ conf = 0.9909 |
| Sea salt (coarse mode) | $5.5 \times 10^{-10}$ $(+0.016, -5.5\times10^{-10})$ | Average value recommended |

For a given relative humidity, $\gamma_{GLYX}$ and $\gamma_{MGLY}$ show weak, nonmonotonic dependence on S:A and S:N, with significant scatter (see Supporting Information). Plotting $\gamma_{GLYX}$ and $\gamma_{MGLY}$ as a function of calculated aerosol pH shows a general positive trend, with $\gamma$ increasing with increasing pH at a given RH. These plots can be found with Supporting Information along with a parameterization of the trend. We note that aerosol pH is not an independent parameter in our calculations, but rather a calculated output of ISORROPIA II as a function of input aerosol composition and RH. Furthermore, the mass transfer and reactive loss processes considered here are not explicitly pH dependent. Therefore, we interpret this apparent dependence on pH to be reflective of pH being a proxy for variations in solute concentration in the aerosol with varying aerosol liquid water content. All variations in $\gamma_{GLYX}$ and $\gamma_{MGLY}$ as a result of varying SNA aerosol composition or pH are incorporated in the error bars shown in the top panels of Figures 1 and 2, and therefore accounted for in the weighted least squares fits, and the uncertainty ranges provided in the parameterizations in Tables 4 and 5.

## 4 Atmospheric Implications

We present revised recommendations for the reactive uptake coefficient for glyoxal and methylglyoxal for several cloud and aerosol types. The values we calculated under many conditions are lower than those currently used in large-scale models such as GEOS-Chem, although we note that the parameterization presented in Table 4 at the experimental conditions of Liggio et

al. (2005), 55% RH, yields $\gamma_{GLY} = 3.6 \times 10^{-3}$, which is within 24% of their experimental value ($2.9 \times 10^{-3}$). We expect application of these parameterizations will result in a decrease in the calculated contribution of MGLY uptake to aqueous SOA formation and better representation of spatial variability in aqSOA formed from glyoxal. The reduced contribution of MGLY to aqueous SOA formation due to salting out is consistent with the calculations of Sareen et al. (2017).

5       Reactive uptake of glyoxal and methylglyoxal to other hygroscopic aerosols such as organics is possible, although given the importance of salting effects on this chemistry, and the low expected [OH] concentration in organic aerosols (McNeill, 2015), we expect the contribution of these processes to aqSOA formation to be minor.

      This representation of aqueous SOA formation by GLYX and MGLY, with the treatment of Henry's constants described here, does not take into account the contribution of reversible uptake of GLYX, which could be a significant,

although transient, source of aerosol mass under some conditions (McNeill et al., 2012; Woo and McNeill, 2015). The use of this parameterization together with simpleGAMMA (Woo and McNeill, 2015) would give representation of both aqSOA formation types by GLYX.

**Acknowledgements**

The authors acknowledge NSF for funding this work (Award AGS-1546136). We are grateful to Tzung-May Fu for helpful discussions.

**Supplementary Information**

      Details of the GEOS-Chem v11 cloud and aerosol specifications, plots of $\gamma$ as a function of aerosol composition and

calculated aerosol pH, a parameterization for $\gamma$ as a function of calculated aerosol pH, and the MATLAB routine for calculating $\gamma$ can be found in the supplementary information.

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
