# Peer review of "Technical note: Updated parameterization of the reactive uptake of glyoxal and methylglyoxal by atmospheric aerosols and cloud droplets"

_Atmospheric Chemistry and Physics, 2018_

## Referee Comment (RC1) · Anonymous Referee #1 · 16 Feb 2018

**General comments:**
Curry et al. present updated values for reactive uptake of glyoxal and methylgloxal for use in models. These are based on RH parameterizations of maritime and remote aerosols and clouds that are comprised of sodium chloride or mixtures of ammonium nitrate and ammonium sulfate. These parameterizations are based on recent (bulk and aerosol) laboratory data that have probed gas-particle partitioning as a function of pH and aerosol or bulk solution chemical composition. This work will result in a more accurate reflection of SOA formed from the water-soluble aldehydes glyoxal and methylglyoxal. I have a few comments, but recommend publication in ACP after minor changes.

**Specific comments:**
1. Methods and Data: Statistical analysis and parameter estimation: It should be clarified in this section (rather than 3.2) that the authors also tested pH and aerosol chemical composition and the results of those are in the SI.

2. Methods and Data: Calculating the Henry's constant: clarify that the $K_{H,w}$ values are the effective Henry's law constants that incorporate hydration.

3. Section 3.2: It looks as though there is a strong correlation of $\gamma_{GLY}$ with pH in Fig. S1 so it is not clear how the authors determined that pH is not necessary.

4. Figure 1 and Figure 2: The "average $\gamma_{GLYX}$" and "average $\gamma_{MGLY}$" values are recommended due to large scatter and lack of correlation with RH. What "average" are the authors referring to? The black line does not appear to be the mean of the $\gamma_{GLYX}$ or $\gamma_{MGLY}$ values.

**Minor comments:**
5. Introduction: Work by David De Haan's group should also be cited for brown carbon formation, e.g.: Powelson et al. (2014) ES&T "Brown carbon formation by aqueous-phase aldehyde reactions with amines and ammonium sulfate"

6. Atmospheric Implications: Consider citing Sareen et al. (2017) ES&T, "Potential of Aerosol Liquid Water to Facilitate Organic Aerosol Formation: Assessing Knowledge Gaps about Precursors and Partitioning" as salting constants were included in this work and they do find that methylglyoxal is a minor contributor as Curry et al. predict.

7. Figure S1: consider changing y-axis to $\gamma_{GLYX}$ for consistency with the main text.

8. References: check for formatting issues (e.g. "Henry ' s Law" in Aster et al., "ChemPhysChem" in Herrmann et al., "(NH4(+))" in Nozière et al., etc.)

---

## Referee Comment (RC2) · Anonymous Referee #2 · 17 Mar 2018

This technical note describes the calculation of uptake coefficients for glyoxal and methylglyoxal based on measured values for Henry's Law coefficients as a function of various salt concentrations and on modeled values for OH radical concentrations in cloudwater and aqueous aerosol. These uptake coefficients are sorely needed, and this note is sure to be of value to the field. The results and methods are clearly described and presented. The authors at one point compare their calculated uptake coefficients for glyoxal on SNA aerosol at 50% RH to laboratory measurements on ammonium sulfate aerosol at 55% RH by Liggio et al.(1). The new, calculated results are high by a factor of three, which seems like reasonable agreement in this field. However, values used for certain parameters are not given, and I have two concerns about the

scope of the conclusions.

Specific Comments

This study appears to take into account only one kind of irreversible reactivity: oxidation by dissolved OH radicals. Can the authors justify that this reaction is more important than all other irreversible aqueous-phase reactions involving dissolved dicarbonyls, such as organosulfate formation, or the non-radical reactions with ammonium sulfate that the authors have studied in the past? I think it is unlikely that using effective Henry's law coefficients, even ones that include salting in / salting out effects, accounts for all of these processes, and the authors allude to this problem in the final paragraph. Given this problem, could the authors be getting reasonably accurate results for glyoxal uptake for the wrong reasons (due to a second error pushing the results in the opposite direction of the first)? It would be helpful to discuss this limitation and the magnitude of the uncertainties more thoroughly to help readers better interpret the results.

Second, the authors have chosen to ignore the effects of sulfate / nitrate ratios, sulfate / ammonium ratios, and pH on glyoxal uptake coefficients and focus exclusively on the effects of relative humidity. Based on Figure S1, I acknowledge that RH appears to be more important than these other three factors. However, Figure S1 shows that sulfate / nitrate ratios, sulfate / ammonium ratios, and pH all have non-linear effects on glyoxal uptake that are as large as the effects of increasing the RH from 70 to 99%. In addition, the laboratory experiments of Liggio et al.[1] showed that glyoxal uptake coefficients depend on aerosol acidity. Just because these effects are non-linear does not mean that they can or should be ignored. In the manuscript, the single statement on p. 5 line 18 that discusses Figure S1 ("no clear correlation is apparent") is at best an oversimplification, and might even be seen as misleading.

Technical comments

While the reported parameter values seem reasonable, I was unable to find the values used for two key parameters: the accommodation coefficient (alpha) and the aqueous

diffusion coefficients. Are the terms that include these parameters not very influential on the overall values calculated for the uptake coefficients?

Abstract (line 12): I think that the statement "We take into account ... aqueous-phase chemical kinetics" should be modified given the first concern described above. Only the chemical kinetics of oxidation reactions with OH are taken into account in this study, not the chemical kinetics of other irreversible reactions.

Reference cited

1. Liggio, J.; Li, S.-M.; McLaren, R., Reactive uptake of glyoxal by particulate matter. J. Geophys. Res. 2005, 110, D10304. doi:10.1029/2004JD005113

---

## Author Comment (AC2) · 8 Jun 2018

**Reviewer comments are copied below. Our responses are written below each comment in blue font.**

*REVIEWER 2*

This technical note describes the calculation of uptake coefficients for glyoxal and methylglyoxal based on measured values for Henry's Law coefficients as a function of various salt concentrations and on modeled values for OH radical concentrations in cloudwater and aqueous aerosol. These uptake coefficients are sorely needed, and this note is sure to be of value to the field. The results and methods are clearly described and presented.

We thank the reviewer for this overall positive assessment.

The authors at one point compare their calculated uptake coefficients for glyoxal on SNA aerosol at 50% RH to laboratory measurements on ammonium sulfate aerosol at 55% RH by Liggio et al.(1). The new, calculated results are high by a factor of three, which seems like reasonable agreement in this field.

We actually had not done the comparison mentioned by the reviewer, and it is an interesting suggestion. Since the dependence on relative humidity is strong near 50% RH (see Figure 1 top panel), if you use our SNA parameterization for glyoxal from Table 4 and plug in 55% RH, the result is $\gamma_{GLYX}$ = $3.6\times10^{-3}$, which is only 24% higher than the experimental value of Liggio et al. We now mention this in the Atmospheric Implications section: "The values we calculated under many conditions are lower than those currently used in large-scale models such as GEOS-Chem. However, we note that the parameterization presented in Table 4 at the experimental conditions of Liggio et al. (2005), 55% RH, yields $\gamma_{GLY}$ = $3.6\times10^{-3}$, which is within 24% of their experimental value ($2.9\times10^{-3}$)."

However, values used for certain parameters are not given, and I have two concerns about the scope of the conclusions.

Specific Comments

This study appears to take into account only one kind of irreversible reactivity: oxidation by dissolved OH radicals. Can the authors justify that this reaction is more important than all other irreversible aqueous-phase reactions involving dissolved dicarbonyls, such as organosulfate formation, or the non-radical reactions with ammonium sulfate that the authors have studied in the past? I think it is unlikely that using effective Henry's law coefficients, even ones that include salting in / salting out effects, accounts for all of these processes, and the authors allude to this problem in the final paragraph. Given this problem, could the authors be getting reasonably accurate results for glyoxal uptake for the wrong reasons (due to a second error pushing the results in the opposite direction of the first)? It would be helpful to discuss this limitation and the magnitude of the uncertainties more thoroughly to help readers better interpret the results.

We chose to focus on reactive uptake driven by OH oxidation since this is the dominant irreversible loss process for GLYX and MGLY in aqueous aerosols and cloud droplets. This reaction is the initiation step for most radical-based chemistry of GLYX and MGLY in the aqueous phase, including organic acid formation and organosulfate formation (McNeill et al., 2012; Perri et al., 2010). Other irreversible loss processes such as imidazole formation occur at much longer timescales (Teich et al., 2016; Yu et al., 2011). The consistency of our calculations with the experimental reactive uptake coefficient of Liggio at

al., a system in which the multiple aqueous-phase processes mentioned here were active, provides support for our approach.  We have expanded our discussion of this matter in the text.

Finally, as we note in the final paragraph of the manuscript, "This representation of aqueous SOA formation by GLYX and MGLY, with the treatment of Henry's constants described here, does not take into account the contribution of reversible uptake of GLYX, which could be a significant, although transient, source of aerosol mass under some conditions (McNeill et al., 2012; Woo and McNeill, 2015). The use of this parameterization together with simpleGAMMA (Woo and McNeill, 2015) would give representation of both aqSOA formation types by GLYX."

Second, the authors have chosen to ignore the effects of sulfate / nitrate ratios, sulfate / ammonium ratios, and pH on glyoxal uptake coefficients and focus exclusively on the effects of relative humidity. Based on Figure S1, I acknowledge that RH appears to be more important than these other three factors. However, Figure S1 shows that sulfate / nitrate ratios, sulfate / ammonium ratios, and pH all have non-linear effects on glyoxal uptake that are as large as the effects of increasing the RH from 70 to 99%. In addition, the laboratory experiments of Liggio et al.[1] showed that glyoxal uptake coefficients depend on aerosol acidity. Just because these effects are non-linear does not mean that they can or should be ignored. In the manuscript, the single statement on p. 5 line 18 that discusses Figure S1 ("no clear correlation is apparent") is at best an oversimplification, and might even be seen as misleading.

We agree with the reviewer that more discussion of the dependence of $\gamma$GLYX and $\gamma$MGLY on pH, S:N, and S:A is required. In this study, variation in aerosol pH was not controlled independently but rather it developed from the variation in S:N and S:A. Since the correlation with S:N and S:A was not strong, and aerosol pH is not a variable in GEOS-Chem, we chose not to address this dependence in the original manuscript. However, since we see the value of providing this information despite its lack of direct applicability to GEOS-Chem simulations, and we agree with the reviewer that not doing so may lead to misunderstanding, we now discuss this data in more detail and include parameterizations for $\gamma$GLYX and $\gamma$MGLY as functions of pH.

The dependence of $\gamma$GLYX and $\gamma$MGLY on S:N and S:A is, as the reviewer noted, weak compared to the dependence on RH and pH.  Furthermore, the plots in Figure S1 show scatter and a lack of monotonicity in the dependence that suggest a lack of a mechanistic basis for the variation observed. As a result, we choose not to include these variables in the parameterization. The variation as a function of S:N and S:A is included in the error bars on the RH fits. We now discuss this in more detail in section 3.

Technical comments

While the reported parameter values seem reasonable, I was unable to find the values used for two key parameters: the accommodation coefficient (alpha) and the aqueous diffusion coefficients. Are the terms that include these parameters not very influential on the overall values calculated for the uptake coefficients?

Thank you for catching this omission, we have now included this information in section 2. The aqueous-phase diffusion coefficient used for glyoxal was $D_{aq} = 10^{-9}$ m$^2$/s and the accommodation coefficient used was $\alpha = 0.02$. $D_{aq}$ does not vary much for small species, and this value is typical for small organics (Bird et al., 2006). This value of $\alpha$ is an estimate based assuming that $\alpha$ for GLY and MGLY are similar to that

of formaldehyde uptake to water (Jayne et al., 1992). As the reviewer suggests, the calculation is insensitive to within 10% for a 50% variation in $\alpha$ for values of $\gamma < 10^{-3}$.

Abstract (line 12): I think that the statement "We take into account . . . aqueous-phase chemical kinetics" should be modified given the first concern described above. Only the chemical kinetics of oxidation reactions with OH are taken into account in this study, not the chemical kinetics of other irreversible reactions.

In response to this suggestion, we have modified this sentence to be more specific and accurate, replacing "aqueous-phase chemical kinetics" with "irreversible reaction of the organic species with OH in the aqueous phase"

*References*

Bird, R. B., Stewart, W. E. and Lightfoot, E. N.: Transport Phenomena, Revised 2nd Edition, John Wiley & Sons, Inc., 2006.

Jayne, J. T., Duan, S. X., Davidovits, P., Worsnop, D. R., Zahniser, M. S. and Kolb, C. E.: Uptake of gas-phase aldehydes by water surfaces, J. Phys. Chem., 96(13), 5452–5460, doi:10.1021/j100192a049, 1992.

McNeill, V. F., Woo, J. L., Kim, D. D., Schwier, A. N., Wannell, N. J., Sumner, A. J. and Barakat, J. M.: Aqueous-Phase Secondary Organic Aerosol and Organosulfate Formation in Atmospheric Aerosols: A Modeling Study, Environ. Sci. Technol., 46(15), 8075–8081, doi:10.1021/es3002986, 2012.

Perri, M. J., Lim, Y. B., Seitzinger, S. P. and Turpin, B. J.: Organosulfates from glycolaldehyde in aqueous aerosols and clouds: Laboratory studies, Atmos. Environ., 44(21–22), 2658–2664, doi:10.1016/j.atmosenv.2010.03.031, 2010.

Sareen, N., Waxman, E. M., Turpin, B. J., Volkamer, R. and Carlton, A. M. G.: Potential of aerosol liquid water to facilitate organic aerosol formation: assessing knowledge gaps about precursors and partitioning, Environ. Sci. Technol., acs.est.6b04540, doi:10.1021/acs.est.6b04540, 2017.

Teich, M., van Pinxteren, D., Kecorius, S., Wang, Z. and Herrmann, H.: First quantification of imidazoles in ambient aerosol particles: Potential photosensitizers, brown carbon constituents and hazardous components, Environ. Sci. Technol., 50(3), acs.est.5b05474, doi:10.1021/acs.est.5b05474, 2016.

Woo, J. L. and McNeill, V. F.: simpleGAMMA v1.0 – a reduced model of secondary organic aerosol formation in the aqueous aerosol phase (aaSOA), Geosci. Model Dev., 8(6), 1821–1829, doi:10.5194/gmd-8-1821-2015, 2015.

Yu, G., Bayer, A. R., Galloway, M. M., Korshavn, K. J., Fry, C. G. and Keutsch, F. N.: Glyoxal in aqueous ammonium sulfate solutions: products, kinetics and hydration effects., Environ. Sci. Technol., 45(15), 6336–6342, doi:10.1021/es200989n, 2011.

---

## Author Response (AR1)

**COLUMBIA UNIVERSITY**
**IN THE CITY OF NEW YORK**

**DEPARTMENT OF CHEMICAL ENGINEERING**

Prof. Manabu Shiraiwa
Department of Chemistry
University of California, Irvine
Irvine, CA 92697

June 19, 2018

Dear Manabu,

We thank you for the opportunity to respond to the reviewers' comments and submit a revised version of our manuscript acp-2018-51, entitled "Technical note: Updated parameterization of the reactive uptake of glyoxal and methylglyoxal by atmospheric aerosols and cloud droplets." We have revised the manuscript based on the reviewers' comments and we feel that it has become stronger as a result. Our conclusions remain the same.

The most significant change to the manuscript, which was requested by both reviewers, is that we have expanded our discussion of the dependence on the reactive uptake coefficients on particle composition, and the apparent dependence on calculated aerosol pH. Aerosol pH is not an independent parameter in our calculations, but rather a calculated output of ISORROPIA II as a function of input aerosol composition and RH. Furthermore, the mass transfer and reactive loss processes considered in our calculations are not explicitly pH dependent. Therefore, we interpret this apparent dependence on pH to be reflective of pH being a proxy for variations in solute concentration in the aerosol with varying aerosol liquid water content. For these reasons, we changed the wording in the abstract and elsewhere in the manuscript where it was previously said that $\gamma_{GLYX}$ and $\gamma_{MGLY}$ were calculated as a function of pH since this is not strictly correct, and may be misleading for readers. We include a parameterization for the dependence of $\gamma_{GLYX}$ and $\gamma_{MGLY}$ on pH at the request of the reviewers, but we have kept it in the Supporting Information. We note that all variations in $\gamma_{GLYX}$ and $\gamma_{MGLY}$ as a result of varying SNA aerosol composition or pH are incorporated in the error bars shown in the top panels of Figures 1 and 2, and therefore accounted for in the weighted least squares fits, and the uncertainty ranges provided in the parameterizations in Tables 4 and 5.

We attach our point-by-point response to the reviewers' comments, along with versions of the revised manuscript and supporting information with changes highlighted. I note that, since the manuscript was finalized after the responses to the reviewers were uploaded on 6/7/2018, there are some minor differences between the final revised manuscript and the descriptions of the changes made to the manuscript in the response documents.

We hope that you find this revised manuscript suitable for publication in *ACP*.

Best Regards,

*Faye*

Prof. V. Faye McNeill (vfm2103@columbia.edu)

816 S. W. Mudd      Mail Code 4721      500 West 120th Street      New York, NY 10027      212-854-2869      Fax 212-854-3054

**Reviewer comments are copied below. Our responses are written below each comment in blue font.**

*REVIEWER 1*

General comments: Curry et al. present updated values for reactive uptake of glyoxal and methylgloxal for use in models. These are based on RH parameterizations of maritime and remote aerosols and clouds that are comprised of sodium chloride or mixtures of ammonium nitrate and ammonium sulfate. These parameterizations are based on recent (bulk and aerosol) laboratory data that have probed gas-particle partitioning as a function of pH and aerosol or bulk solution chemical composition. This work will result in a more accurate reflection of SOA formed from the water-soluble aldehydes glyoxal and methylglyoxal. I have a few comments, but recommend publication in ACP after minor changes.

We thank the reviewer for the overall positive comments.

Specific comments:

1. Methods and Data: Statistical analysis and parameter estimation: It should be clarified in this section (rather than 3.2) that the authors also tested pH and aerosol chemical composition and the results of those are in the SI.

Thanks for this suggestion. The title of the subsection "Particle types" has been modified to "Particle types and composition" and we have inserted the sentence "Calculated results for $\gamma_{GLY}$ and $\gamma_{MGLY}$ as a function of S:A and S:N are available in the Supporting Information" after the sentence describing the particle types and composition and RH ranges considered. As discussed below, the results as a function of pH are now discussed separately in section 3.

2. Methods and Data: Calculating the Henry's constant: clarify that the $K_{H,w}$ values are the effective Henry's law constants that incorporate hydration.

The sentence "Note that the $K_H$ values are effective Henry's constants which account for hydration of the carbonyl species upon uptake." has been added after the introduction of $K_H$ in equation (3).

3. Section 3.2: It looks as though there is a strong correlation of $\gamma$GLY with pH in Fig. S1 so it is not clear how the authors determined that pH is not necessary.

The reviewer is correct, there is a correlation of $\gamma$GLYX with pH in figure S1. In this study, variation in aerosol pH was not controlled independently, but rather it developed from the variation in S:N and S:A. Since the correlation with S:N and S:A was not strong, and aerosol pH is not a variable in GEOS-Chem, we chose not to address this dependence in the original manuscript. Since we see the value of providing this information despite its lack of direct applicability to GEOS-Chem simulations, we now discuss this data in more detail and include parameterizations for $\gamma$GLYX and $\gamma$MGLY as functions of pH.

4. Figure 1 and Figure 2: The "average $\gamma$GLYX" and "average $\gamma$MGLY" values are recommended due to large scatter and lack of correlation with RH. What "average" are the authors referring to? The black line does not appear to be the mean of the $\gamma$GLYX or $\gamma$MGLY values.

The statistical analysis and parameter estimation is described in section 2. The averages were obtained via weighted linear least squares regression, so points with lower uncertainty were given more weight in

the average, and the average does not lie in the middle of all of the points. We have added a parenthetical note to make this more clear: "The scatter in the calculated $\gamma_{GLYX}$ led to a low-confidence result from the weighted least squares regression. For this reason, we recommend use of the average $\gamma_{GLYX}$ value (obtained via weighted linear least squares regression, with slope = 0) in lieu of a parameterization (Table 4)."

Minor comments:

5. Introduction: Work by David De Haan's group should also be cited for brown carbon formation, e.g.: Powelson et al. (2014) ES&T "Brown carbon formation by aqueous-phase aldehyde reactions with amines and ammonium sulfate"

This reference, as well as a reference to De Haan et al. ES&T 52 (7) 4061-4071 (2018) have been added.

6. Atmospheric Implications: Consider citing Sareen et al. (2017) ES&T, "Potential of Aerosol Liquid Water to Facilitate Organic Aerosol Formation: Assessing Knowledge Gaps about Precursors and Partitioning" as salting constants were included in this work and they do find that methylglyoxal is a minor contributor as Curry et al. predict.

We now mention and cite this study: "The reduced contribution of MGLY to aqueous SOA formation due to salting out is consistent with the calculations of Sareen et al. (2017)."

7. Figure S1: consider changing y-axis to $\gamma$GLYX for consistency with the main text.

The change has been made.

8. References: check for formatting issues (e.g. "Henry ' s Law" in Aster et al., "ChemPhysChem" in Herrmann et al., "(NH4(+))" in Nozière et al., etc.)

We have checked the reference list and made corrections where necessary. ChemPhysChem is the correct name of the journal in the Herrmann et al. (2010) reference.

**Reviewer comments are copied below. Our responses are written below each comment in blue font.**

*REVIEWER 2*

This technical note describes the calculation of uptake coefficients for glyoxal and methylglyoxal based on measured values for Henry's Law coefficients as a function of various salt concentrations and on modeled values for OH radical concentrations in cloudwater and aqueous aerosol. These uptake coefficients are sorely needed, and this note is sure to be of value to the field. The results and methods are clearly described and presented.

We thank the reviewer for this overall positive assessment.

The authors at one point compare their calculated uptake coefficients for glyoxal on SNA aerosol at 50% RH to laboratory measurements on ammonium sulfate aerosol at 55% RH by Liggio et al.(1). The new, calculated results are high by a factor of three, which seems like reasonable agreement in this field.

We actually had not done the comparison mentioned by the reviewer, and it is an interesting suggestion. Since the dependence on relative humidity is strong near 50% RH (see Figure 1 top panel), if you use our SNA parameterization for glyoxal from Table 4 and plug in 55% RH, the result is $\gamma_{GLYX}$ = $3.6\times10^{-3}$, which is only 24% higher than the experimental value of Liggio et al. We now mention this in the Atmospheric Implications section: "The values we calculated under many conditions are lower than those currently used in large-scale models such as GEOS-Chem. However, we note that the parameterization presented in Table 4 at the experimental conditions of Liggio et al. (2005), 55% RH, yields $\gamma_{GLY}$ = $3.6\times10^{-3}$, which is within 24% of their experimental value ($2.9\times10^{-3}$)."

However, values used for certain parameters are not given, and I have two concerns about the scope of the conclusions.

Specific Comments

This study appears to take into account only one kind of irreversible reactivity: oxidation by dissolved OH radicals. Can the authors justify that this reaction is more important than all other irreversible aqueous-phase reactions involving dissolved dicarbonyls, such as organosulfate formation, or the non-radical reactions with ammonium sulfate that the authors have studied in the past? I think it is unlikely that using effective Henry's law coefficients, even ones that include salting in / salting out effects, accounts for all of these processes, and the authors allude to this problem in the final paragraph. Given this problem, could the authors be getting reasonably accurate results for glyoxal uptake for the wrong reasons (due to a second error pushing the results in the opposite direction of the first)? It would be helpful to discuss this limitation and the magnitude of the uncertainties more thoroughly to help readers better interpret the results.

We chose to focus on reactive uptake driven by OH oxidation since this is the dominant irreversible loss process for GLYX and MGLY in aqueous aerosols and cloud droplets. This reaction is the initiation step for most radical-based chemistry of GLYX and MGLY in the aqueous phase, including organic acid formation and organosulfate formation (McNeill et al., 2012; Perri et al., 2010). Other irreversible loss processes such as imidazole formation occur at much longer timescales (Teich et al., 2016; Yu et al., 2011). The consistency of our calculations with the experimental reactive uptake coefficient of Liggio at

al., a system in which the multiple aqueous-phase processes mentioned here were active, provides support for our approach.  We have expanded our discussion of this matter in the text.

Finally, as we note in the final paragraph of the manuscript, "This representation of aqueous SOA formation by GLYX and MGLY, with the treatment of Henry's constants described here, does not take into account the contribution of reversible uptake of GLYX, which could be a significant, although transient, source of aerosol mass under some conditions (McNeill et al., 2012; Woo and McNeill, 2015). The use of this parameterization together with simpleGAMMA (Woo and McNeill, 2015) would give representation of both aqSOA formation types by GLYX."

Second, the authors have chosen to ignore the effects of sulfate / nitrate ratios, sulfate / ammonium ratios, and pH on glyoxal uptake coefficients and focus exclusively on the effects of relative humidity. Based on Figure S1, I acknowledge that RH appears to be more important than these other three factors. However, Figure S1 shows that sulfate / nitrate ratios, sulfate / ammonium ratios, and pH all have non-linear effects on glyoxal uptake that are as large as the effects of increasing the RH from 70 to 99%. In addition, the laboratory experiments of Liggio et al.[1] showed that glyoxal uptake coefficients depend on aerosol acidity. Just because these effects are non-linear does not mean that they can or should be ignored. In the manuscript, the single statement on p. 5 line 18 that discusses Figure S1 ("no clear correlation is apparent") is at best an oversimplification, and might even be seen as misleading.

We agree with the reviewer that more discussion of the dependence of $\gamma$GLYX and $\gamma$MGLY on pH, S:N, and S:A is required. In this study, variation in aerosol pH was not controlled independently but rather it developed from the variation in S:N and S:A. Since the correlation with S:N and S:A was not strong, and aerosol pH is not a variable in GEOS-Chem, we chose not to address this dependence in the original manuscript. However, since we see the value of providing this information despite its lack of direct applicability to GEOS-Chem simulations, and we agree with the reviewer that not doing so may lead to misunderstanding, we now discuss this data in more detail and include parameterizations for $\gamma$GLYX and $\gamma$MGLY as functions of pH.

The dependence of $\gamma$GLYX and $\gamma$MGLY on S:N and S:A is, as the reviewer noted, weak compared to the dependence on RH and pH.  Furthermore, the plots in Figure S1 show scatter and a lack of monotonicity in the dependence that suggest a lack of a mechanistic basis for the variation observed. As a result, we choose not to include these variables in the parameterization. The variation as a function of S:N and S:A is included in the error bars on the RH fits. We now discuss this in more detail in section 3.

Technical comments

While the reported parameter values seem reasonable, I was unable to find the values used for two key parameters: the accommodation coefficient (alpha) and the aqueous diffusion coefficients. Are the terms that include these parameters not very influential on the overall values calculated for the uptake coefficients?

Thank you for catching this omission, we have now included this information in section 2. The aqueous-phase diffusion coefficient used for glyoxal was $D_{aq}$ = 10$^{-9}$ m$^2$/s and the accommodation coefficient used was $\alpha$ = 0.02. $D_{aq}$ does not vary much for small species, and this value is typical for small organics (Bird et al., 2006). This value of $\alpha$ is an estimate based assuming that $\alpha$ for GLY and MGLY are similar to that

of formaldehyde uptake to water (Jayne et al., 1992). As the reviewer suggests, the calculation is insensitive to within 10% for a 50% variation in $\alpha$ for values of $\gamma < 10^{-3}$.

Abstract (line 12): I think that the statement "We take into account . . . aqueous-phase chemical kinetics" should be modified given the first concern described above. Only the chemical kinetics of oxidation reactions with OH are taken into account in this study, not the chemical kinetics of other irreversible reactions.

In response to this suggestion, we have modified this sentence to be more specific and accurate, replacing "aqueous-phase chemical kinetics" with "
[revised manuscript text omitted]

[Figure]

**Figure S1.** Calculated $\gamma_{GLYX}$ as a function of aerosol composition for SNA aerosols at varying relative humidity.

[Figure]

**Figure S2**. Calculated $\gamma_{GLYX}$ and $\gamma_{MGLY}$ for SNA aerosols displayed as a function of calculated aerosol pH and varying relative humidity

**Table S2.** Recommended parameterization of $\gamma_{GLYX}$ as a function of RH and pH. For pH $\leq$ pH$_{break}$, $\gamma_{GLYX}$ = exp($a+b*$pH). For pH $>$ pH$_{break}$, $\gamma_{GLYX}$ = $c+d*$pH. Parameterization valid for $-1.05 \leq$ pH $\leq 4.64$.

| RH | pH$_{break}$ | $a$ | $b$ | $c$ | $d$ |
|---|---|---|---|---|---|
| 50% | 2.30 | * | * | $1.36(\pm 0.21)\times10^{-3}$ | $4.31(\pm 0.07)\times10^{-3}$ |
| 70% | 2.50 | $-9.8 \pm 0.2$ | $1.37 \pm 0.05$ | $-1.53(\pm 0.11)\times10^{-3}$ | $7.02(\pm 0.34)\times10^{-4}$ |
| 80% | 2.63 | $-12.2 \pm 0.3$ | $1.58 \pm 0.13$ | $-2.65(\pm 0.19)\times10^{-4}$ | $1.12(\pm 0.06)\times10^{-4}$ |
| 90% | 2.86 | $-15.6 \pm 0.6$ | $2.37 \pm 0.36$ | $-2.81(\pm 0.20)\times10^{-5}$ | $1.06(\pm 0.06)\times10^{-5}$ |
| 95% | 3.13 | $-14.8 \pm 1.0$ | $1.84 \pm 0.51$ | $-3.51(\pm 0.23)\times10^{-5}$ | $1.21(\pm 0.06)\times10^{-5}$ |
| 99% | 3.69 | $-14.7 \pm 1.1$ | $1.90 \pm 0.48$ | $-8.34(\pm 0.41)\times10^{-5}$ | $2.39(\pm 0.09)\times10^{-5}$ |

\* for 50% RH, pH $\leq$ pH$_{break}$, $\gamma_{GLYX}$ = $a+b*$pH$+c*$pH$^2$, where a = $3.0(\pm 0.5)\times10^{-3}$, b = $-7.4(\pm 4.5)\times10^{-4}$, and c = $2.3(\pm 0.3)\times10^{-3}$

**Table S3.** Recommended parameterization of $\gamma_{MGLY}$ as a function of RH and pH. For pH $\leq$ pH$_{break}$, $\gamma_{MGLY}$ = exp($a+b*$pH) for pH $>$ pH$_{break}$, $\gamma_{MGLY}$ = $c+d*$pH. Parameterization valid for $-1.05 \leq$ pH $\leq 4.64$.

| RH | pH$_{break}$ | $a$ | $b$ | $c$ | $d$ |
|---|---|---|---|---|---|
| 50% | 2.30 | $-25.7 \pm 0.2$ | $1.49 \pm 0.13$ | $-5.88(\pm 0.05)\times10^{-10}$ | $2.86(\pm 0.16)\times10^{-10}$ |
| 70% | 2.50 | $-22.7 \pm 0.1$ | $1.37 \pm 0.05$ | $-1.95(\pm 0.13)\times10^{-8}$ | $8.60(\pm 0.41)\times10^{-9}$ |
| 80% | 2.63 | $-22.0 \pm 0.2$ | $1.58 \pm 0.13$ | $-5.30(\pm 0.36)\times10^{-8}$ | $2.21(\pm 0.11)\times10^{-8}$ |
| 90% | 2.86 | $-22.2 \pm 0.6$ | $2.35 \pm 0.35$ | $-5.71(\pm 0.40)\times10^{-8}$ | $2.15(\pm 0.11)\times10^{-8}$ |
| 95% | 3.13 | $-21.3 \pm 1.0$ | $2.04 \pm 0.50$ | $-1.00(\pm 0.07)\times10^{-7}$ | $3.46(\pm 0.17)\times10^{-8}$ |

| 99% | 3.69 | $-19.9 \pm 1.1$ | $1.96 \pm 0.48$ | $-5.07(\pm 0.32) \times 10^{-7}$ | $1.46(\pm 0.07) \times 10^{-7}$ |

**MATLAB routine for calculating reactive uptake coefficients**

Example shown for glyoxal uptake to maritime clouds.

```matlab
clear;

Da = 1e-9; %Gly aqueous diffusion constant, m2/s
kOH = 1.1e9; %Gly-OH bimolecular aqueous rate constant, 1/M/s
kB = 1.38e-23; %Boltzmann constant, m^2*kg/s^2/K
T = 298; % Temp, K
M = 58.04/6.023e23/1000; %% mass of one GLY molecule, kg
w = sqrt(8*kB*T/pi()/M); %m/s
alpha = 0.02;

%% Maritime CLOUD

    H = 3.6e5; %Henry's constant for dilute conditions, M/atm
    R = 10e-6; %radius, m
    OHconc = [2e-12, 5.3e-12, 3.8e-14]; % Molar

for i = 1:3
    kI=kOH*OHconc(i); % calculate psuedo first order rate constant, 1/s
    q = R*sqrt(kI/Da);
    f = coth(q)-1/q;
    gMC(i) = (1/alpha+w/(4*H*82.06/1000*T*sqrt(Da*kI))*1/f)^-1
end
```